# A Review of Current and Historical Research Contributions to the Development of Ground Autonomous Vehicles for Agriculture

**Valda Rondelli [1,*], Bruno Franceschetti [1] and Dario Mengoli [2]**

1   Department of Agricultural and Food Sciences, University of Bologna, Viale G. Fanin 50, 40127 Bologna, Italy; bruno.franceschetti@unibo.it
2   Department of Electrical, Electronic, and Information Engineering "Guglielmo Marconi", University of Bologna, Viale del Risorgimento 2, 40136 Bologna, Italy; dario.mengoli2@unibo.it
*   Correspondence: valda.rondelli@unibo.it

**Abstract:** In this study, a comprehensive overview of the available autonomous ground platforms developed by universities and research groups that were specifically designed to handle agricultural tasks was performed. As cost reduction and safety improvements are two of the most critical aspects for farmers, the development of autonomous vehicles can be of major interest, especially for those applications that are lacking in terms of mechanization improvements. This review aimed to provide a literature evaluation of present and historical research contributions toward designing and prototyping agricultural ground unmanned vehicles. The review was motivated by the intent to disseminate to the scientific community the main features of the autonomous tractor named BOPS-1960, which was conceived in the 1960s at the Alma Mater Studiorum University of Bologna (UNIBO). Jointly, the main characteristics of the modern DEDALO unmanned ground vehicle (UGV) for orchard and vineyard operations that was designed recently were evaluated. The basic principles, technology and sensors used in the two UNIBO prototypes are described in detail, together with an analysis of UGVs for agriculture conceived in recent years by research centers all around the world.

**Keywords:** precision agriculture; UGV; tractor; sensors; unmanned vehicles

## 1. Introduction

Precision agriculture (PA) and autonomous vehicles are nowadays topics of increasing interest for the primary sector, as well as safety measures for the operator in the use of agricultural machines.

Agriculture is indeed one of the sectors that features the most deadly accidents every year [1–3], and tractors overturning is by far the most frequent cause of injury or death of the operator [1–4].

Autonomous/unmanned ground vehicles (AGVs/UGVs) represent a key factor in reducing the number of fatal injuries since they are able to operate without human intervention in real time and decrease driver actuation during normal operation in the field. Beyond serving as a solution to mitigate the risk related to farming activities, UGVs can help in contrasting the scarcity of human labor, both in terms of quantity and quality or required skill. Other strengths of UGVs can be recognized in terms of the timeliness of the execution and the precision and repeatability of the operations. Furthermore, an autonomous vehicle potentially operates for a longer time without being limited by safety regulations that are addressed to preserve human operator health (e.g., hours per working day with regard to daily vibration exposure to whole-body vibration at work). Safety is a primary issue for tasks involving agricultural machines, and in extreme soil conditions, it may be convenient to replace ride-on tractors with autonomous vehicles [5] to prevent the consequences of potential tractor overturns.

The beginning of PA is dated to the 1980s; nevertheless, the first prototypes of autonomous driving vehicles were designed about 30 years beforehand. PA aims toward defining the correct management for the homogeneous areas of the field, i.e., "doing the right thing, in the right place and in the right time" [6]. Both the spatial and temporal variabilities of all the factors affecting the whole crop production process are assessed and managed to improve input efficiency in dynamic crop management. Increasing efficiency means using less while preserving the same final result or obtaining a better output using the same amount of inputs [5,7]. In addition to the technical aspects, even the economic and environmental consequences need to be considered since the possibility of avoiding overlapping during any field operations (such as spraying and spreading) can significantly reduce the quantity of inputs and increase the working capacity, thus leading to a better environmental and economic sustainability.

In PA, the current deep interest in the development and implementation of autonomous driving systems is related to the need for relieving the operator of the two main aspects consequent to the driving action:

- The physical one, whereby using their movements and physical effort, the driver turns the steering wheel and operates levers and pedals to travel along the expected path;
- The mental one, whereby using their intelligence, the operator actuates the implements and needs continuous attention to perform the specific operation with all the maneuvers the task requires.

Indeed, in agriculture, the machine driver typically has two main tasks: driving the vehicle and managing the tools to guarantee an acceptable level of performance; this level has remained substantially unchanged over the decades [8]. While the operator tasks have been unchanged over this time, the design and features of both the tractors and the connected implements have changed greatly and the difficulties in driving and operating them in the field have increased, mainly for the aged operators. All this has motivated the current most advanced approach in PA, i.e., designing UGVs with artificial intelligence.

Since 1960, the UNIBO has been successfully engaged in this challenge. At the time, under the direction of Professor Giuseppe Stefanelli, two young researchers, namely, Pietro Bosi and Luigi Martelli, designed and prototyped the first programmable unmanned tractor, namely, "BOPS-1960" [9]. Details and parts of the autonomous tractor were very well documented in some papers of the time, but these are not readily accessible, mainly because of the language barrier. With the BOPS-1960 being the first unmanned tractor developed in Italy and presented in local exhibitions of field-plowing operations, we considered it appropriate to recap its main features. Nevertheless, the current autonomous ground vehicles available on the market have dimensions, shapes and features that are different relative to the standard or narrow track tractors, and therefore, at least in the European countries, they are not identified as tractors under the current European regulations [10]. Consequently, they are mostly light platforms with a forward speed of less than 6 km/h, equipped with agricultural implements, and can be either specifically designed or commercial ones.

This review aimed to provide a literature evaluation of present and historical research contributions toward the design and prototyping of agricultural ground unmanned vehicles. The idea of the review was motivated by the intent to disseminate to the scientific community the main features of the BOPS-1960 autonomous tractor conceived in the 1960s at the Institute of Agricultural Mechanization (IMA), which has since merged into the Department of Agricultural and Food Sciences (DISTAL) of the Alma Mater Studiorum University of Bologna (UNIBO), as well as describe the main characteristics of the modern UGV designed at the Department of Electrical Electronic and Information Engineering (DEI) of the UNIBO for orchard and vineyard operations. The basic principles, technology and sensors used in the two UNIBO prototypes are described in detail, together with an analysis of UGVs for agriculture that were conceived by research centers all around the world in recent years.

## 2. Research Activity on Autonomous and Robotic Ground Vehicles for Agricultural Tasks

Given the current interest of researchers, manufacturers and stakeholders regarding autonomous vehicles in agriculture, in this section, the aim was to analyze and summarize the main contributions, findings and solutions prototyped by research groups and universities from all over the world.

### 2.1. Outside Europe

The first prototype to address was from Massey University (New Zealand). As described in Scarfe et al. (2009) [11], the prototype is an autonomous robotic four-wheel-drive platform that was designed to handle harvesting tasks in kiwifruit orchards. Fruit picking is achieved by means of four robotic arms mounted on the platform that are able to reach the fruit hanging above the vehicle (Figure 1).

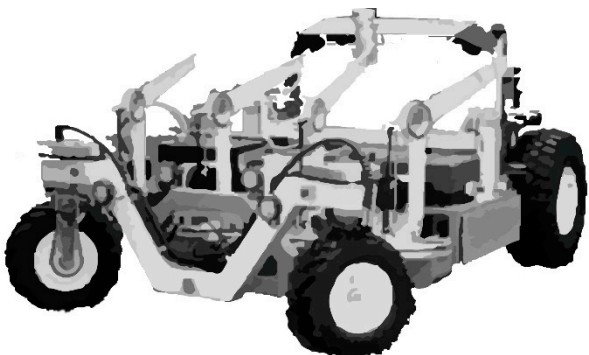

**Figure 1.** Sketch of the autonomous kiwifruit harvester.

This UGV is powered by a combustion engine that acts as a generator to power the hydraulic pump used for locomotion and steering. The arms use electrical motors and the control system algorithms are executed using two commercial dual-processor motherboards. The navigation and localization tasks rely on a differential global positioning system (GPS) receiver in synergy with a digital compass and a computer vision system. The vision system is also used to localize the fruit at harvesting time. The prototype has a total of 10 cameras. Two of them are dedicated to the navigation algorithms to enable obstacle avoidance features, six are used for fruit harvesting and the last two are used to monitor the stored fruits in the bins. The machine can operate continuously by monitoring both its status and the environmental conditions. The first algorithm is used to unload a full bin at the end of the row and automatically collect an empty one and autonomously go to the fuel refill station when needed. The second monitoring algorithm is used to pause the operations when wet conditions are detected and to control floodlights when needed.

Bangkok University (Thailand) developed a three-wheeled platform, with the two rear wheels driven by one 500 W DC motor per wheel (Figure 2). The purpose of the vehicle is the direct sowing of rice on a dried field. This autonomous vehicle consists of several sensors, such as a speed encoder/tachometer and a steering angle sensor, as well as the ability to calculate the overall traveled distance. Autonomous navigation is achieved using a proportional–integral–derivative (PID) controller for speed regulation and a proportional controller to control the steering angle. A GPS antenna and an inertial measurement unit (IMU) are used for the platform localization and attitude to enable waypoints or path following during navigation [12]. All the sensors feed information into an extended Kalman filter (EKF) to allow for precise position estimation of the overall system. The vehicle is able to follow a series of waypoints to enable autonomous seeding application over the field.

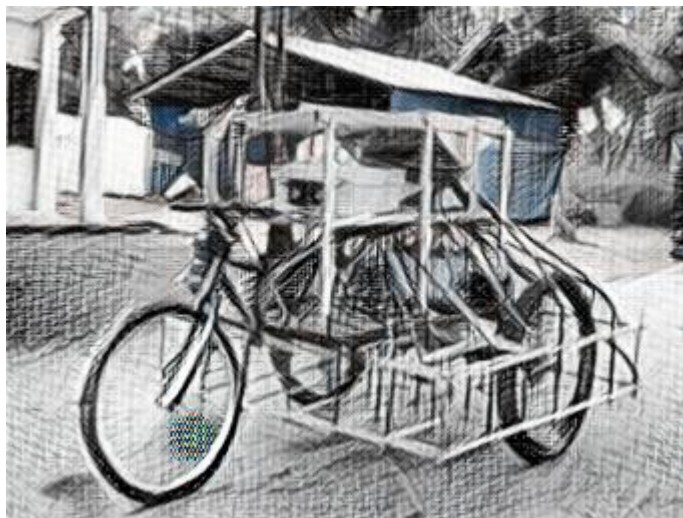

**Figure 2.** Sketch of the autonomous vehicle from Bangkok University.

The Australian Centre for Field Robotics (ACFR) at the University of Sydney designed a UGV platform named "Shrimp". It is a multisensory platform that is able to efficiently detect, trace, localize and map single mango fruits in orchards.

The electric-powered vehicle is equipped with a color camera, a frame with strobo-scope lights, 3D light detection and ranging (LiDAR) and a navigation system that relies on an inertial unit and a global positioning antenna (GPS/INS) (Figure 3) [13]. While autonomously traversing the orchard, the system is able to track and distinguish each fruit without overcounting by means of a multi-view approach. The navigation algorithm seems to rely mostly on the GPS/INS system.

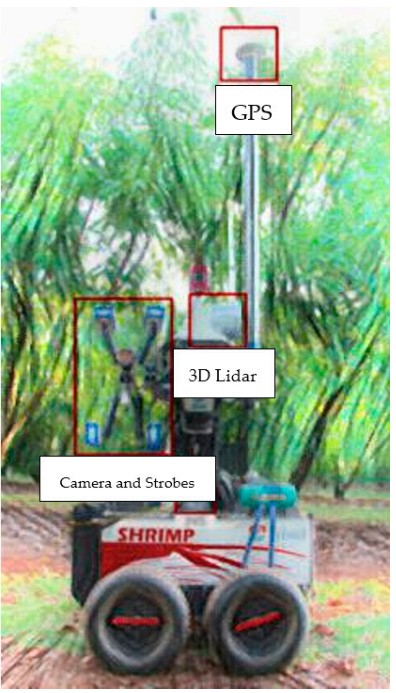

**Figure 3.** Sketch of the robotic platform "Shrimp" that was designed at the University of Sidney.

### 2.2. Within Europe

In Denmark (Aarhus University), a robust horticultural tool carrier named Hortibot, which was derived from an existing commercial machine, was developed by modifying a remote-controlled mower (Figure 4).

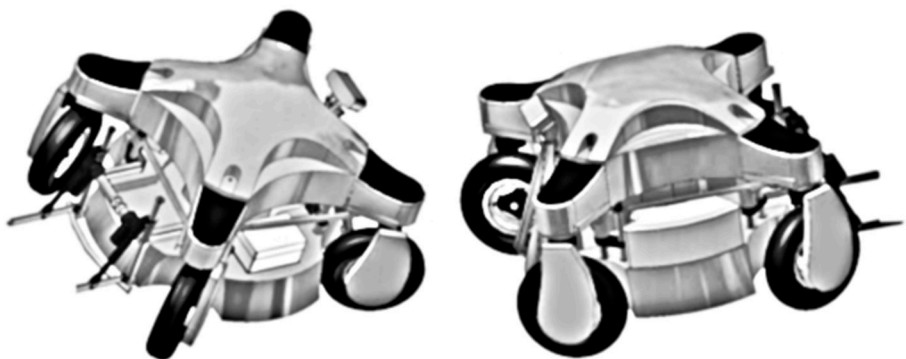

**Figure 4.** Sketch of the Hortibot platform with a 3D row vision system and lift arms.

The HortiBot is able to travel autonomously through several plots with visible rows by using a new commercial row detection system developed by Agrocom Vision (formerly Eco-Dan Inc., Kvistgaard, Denmark), which requires very low use of a GPS [14].

A feasibility study was carried out by Aarhus University and Vitus Bering University College Denmark to evaluate the viability of using the Hortibot mounted with weeding tools in farming operations. The performance was demonstrated through targeted performance that was adapted using knowledge of horticulture. The implemented tools for inter- and intra-row weeding consist of standard A-shaped hoes for inter-row weeding and with bed ridges attached to each end of the implement toolbar. The intra-row weeding is provided by finger and torsion weeders and pneumatic nozzles, which are all attached to five individual units carrying the A-shaped hoes. The pneumatic nozzles are switched on and off by electronically controlled pneumatic valves. Both the navigation control parallel to the rows and within rows are based on computer vision algorithms [15].

Two other research contributions were from Wageningen University (Netherlands). Both were designed for weed control in open fields: one for arable crops, specifically for addressing weed control in maize crops, and the other one was for weeding operations in pastures. The two platforms share a common four-wheel-drive locomotion approach and are powered by a diesel engine.

The first vehicle is able to autonomously control the weeds in a corn field using mechanical weeding elements fixed on the rear part of the vehicle. The vehicle can control the implements by exploiting the navigation data coming from the tractor in the seeding operation (Figure 5) [16]. The UGV autonomous navigation works by leveraging two GPS receivers and a base station for differential GPS (DGPS), and a real-time kinematic (RTK) correction signal is transmitted by means of a radio communication channel [17]. The platform control system is composed of two elements: a high-level controller made of two regulators and a low-level controller with a Smith predictor.

The second vehicle [18] is also equipped with a mechanical vertical axis weeding tool, but in this case, it is mounted on the front of the vehicle. The weeds are detected through computer vision algorithms that analyze the camera images. The camera is placed on a special arm that also holds the GPS antenna (Figure 6). The navigation system is very simple because, as the working scenario is set on meadows or pastures, the vehicle just needs to follow a predefined trajectory. As a safety measure and to ensure proper path following, the vehicle is designed to stop when the GPS signal is lost. Two levels of control are foreseen: a high-level controller is used to manage the navigation and path following, as well as perform the image processing and sending actuators commands; a low-level controller is used to manage sensors and hydraulics.

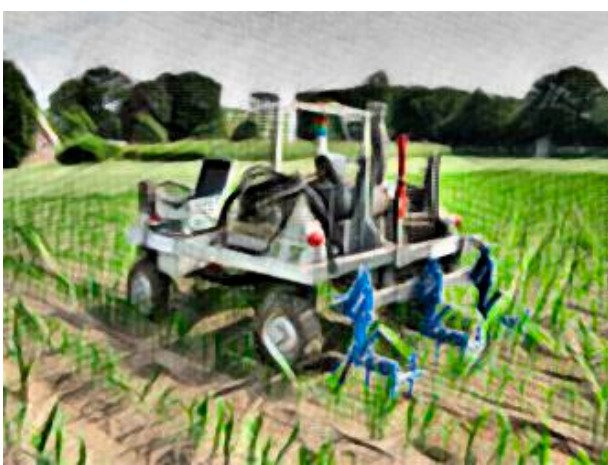

**Figure 5.** Sketch of the robot platform developed by Wageningen University in the field.

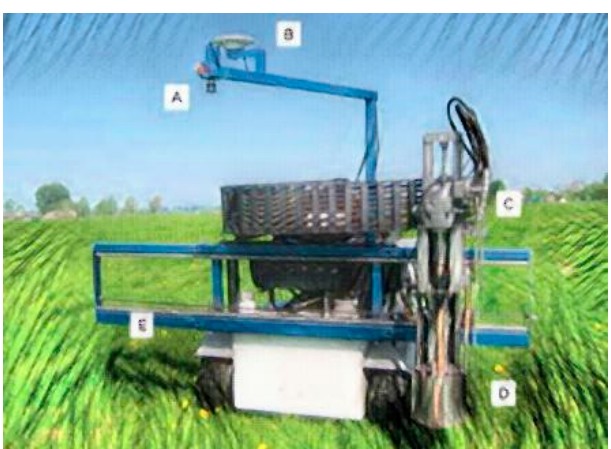

**Figure 6.** Sketch of the second robot vehicle developed by Wageningen University. Visible are (**A**) the camera, (**B**) the GPS antenna, (**C**) the hydraulic motor to actuate the weeder, (**D**) the weeder, and (**E**) the rail to adjust the weeder location.

Aarhus University (Denmark) also developed an autonomous application named GrassBots, which is a UGV that was designed to harvest herbaceous materials intended for biogas plants. The vehicle moves on two hydraulic tracks and is powered by a 74 kW diesel engine. The platform is 3 m wide and equipped with rotary encoders, an IMU and a global navigation satellite system (GNSS) antenna with an RTK localization system. The navigation is powered by open-source software specifically designed for unmanned systems.

GrassBots was designed to operate with very low human intervention and is able to autonomously cover all areas of the field safely and efficiently. A navigation algorithm defines the parallel lines (in terms of a set of waypoints) to follow on the field, its boundaries, obstacles, line widths, driving direction and end-of-field maneuvers [19].

The last UGV discussed here is Armadillo from the University of Southern Denmark, which is a field robotic tool carrier with a modular design to be configurable and adaptable to a wide range of precision agriculture research projects. Armadillo weighs around 425 kg and consists of two 18 × 80 cm footprint track modules, each with an integrated 3.5 kW electric motor, gear and motor controller. The track modules are mounted on the side of an exchangeable tool platform, which allows for an adjustable width and clearance height. The 48 V lithium power pack allows for 10 h of operation [20].

The Armadillo robot supports odometry feedback originating from a brushless motor hall sensor (625 ticks/meter) and is equipped with an IMU and RTK-GNSS receiver connected to the GPSnet.dk reference network.

The Armadillo robot is controlled using an industrial PC (dual-core, 2× GHz) running the open-source ROS-based FroboMind software platform developed by the University of Southern Denmark [21].

### 2.3. DEDALO: The Prototype from the University of Bologna

The Alma Mater Studiorum University of Bologna - Italy has recently developed the DEDALO UGV prototype, which was specifically designed for precision orchard and vineyard management (Figure 7).

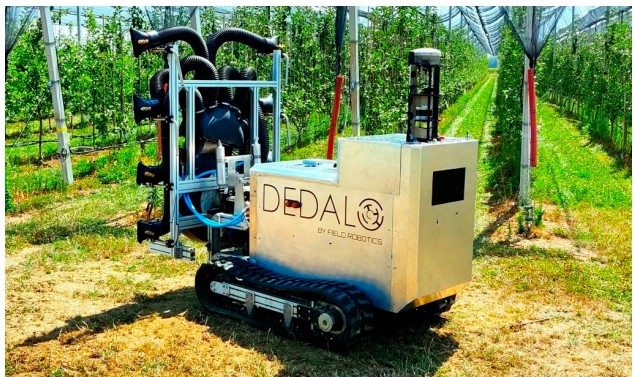

**Figure 7.** DEDALO, which is a UGV that was designed at the UNIBO in the last five years [22].

As reported in Mengoli et al. [22], this prototype is a small and lightweight structure that was designed for spraying and mowing operations in orchards and vineyards. The platform's locomotion system is based on two rubber tracks and driven by electric motors, with the distance between the tracks being adjustable to match the row sizes. The mounted implements are powered by a 16 kW petrol engine due to the quite high energy demand of the attached implements (e.g., the sprayer).

The control logic of the UGV is divided into the following:

- A low-level (LL) controller;
- A high-level (HL) controller.

The LL component is directly connected to the vehicle actuators and motor drivers. It also oversees the safety procedures by transmitting velocity set-points to the electrical motors for locomotion while collecting useful telemetry data. The HL component contains the implementation of the navigation algorithms, and thus, it is the "smartest" element of the system. Using the information provided by the onboard sensors, autonomous navigation is provided in both open-field scenarios and in-row orchard scenarios.

The sensors fitted on the vehicle are as follows:

- Motor encoders for vehicle odometry;
- Nine-axis IMU, with integrated three-axis accelerometers, three-axis gyroscopes and three-axis magnetometers;
- A GNSS receiver for global localization, mainly in the open-field scenario;
- Three-dimensional LiDAR to estimate the surrounding obstacles, mainly for in-row navigation

Two computers (one for the LL control and the second for the HL control) process all the input/output signals and the control algorithms.

The locomotion electrical motors used to drive the platform rubber tracks are equipped with sinusoidal encoders to estimate the speed of the vehicle. An IMU evaluates the orientation of the platform in the 3D space by combining acceleration data, angular velocity and Earth magnetic field readings. To enable autonomous navigation of the platform in any

operating scenario, two complementary elements were installed: the GNSS antenna and LiDAR. The GNSS receiver is a Trimble 8s system, (Raunheim, Germany) that is capable of receiving L1 and L2 signals and using differential correction from an authorized base station in the Trimble network. LiDAR (a 3D laser scanner, Velodyne Lidar Headquarters, San Jose, CA, USA) is used to perceive the environment and to detect the structural characteristics of the orchard and the non-negligible obstacles, such as fences. Two computers are used to split the computational burden of the HL logic from the LL logic, thus maintaining efficient low-level control, which is critical for the motion and the operation of the robot.

The communication and data transfer within the platform can basically be divided into the following:

- Communication between sensors and hardware;
- Communication between motor drivers and input/output (I/O).

To obtain optimal communication between sensors and hardware, a reliable communication channel supporting both the data flow and the low-latency commands to the actuators is used. Conversely, the CANbus protocol is used to connect the I/O subsystem and the motor drivers.

Finally, on the side of the vehicle opposite the implemented linkage system, a screen with a graphical interface was fitted to provide the user with information on the operation performed, the position of the robot and the telemetry data.

## 3. The Early Days in the Research on Autonomous Vehicles for Agriculture

Before 1960, several prototypes of experimental tractors were developed in various countries. The purpose of the early prototypes was mainly to replace the driver's operations by developing autonomous, self-guided or radio-controlled vehicles. The approach taken was to fit sensors and actuators on the tractors available at the time.

The main difficulty in recovering this early information is due to a language barrier because the manuscripts are always in the native language of the researchers, and since they are very old research findings, the papers were not digitally released.

In 1960, Stefanelli [9] summarized the available information in an attempt to collect, as organically as possible, the main research findings from both the Italian and foreign publications, nevertheless claiming that the data reported were not supposed to be exhaustive or complete.

The first study summarized, dated in 1955 from England (Figure 8), involved a Fordson Major tractor connected to a 27.12 MHz transmitter that was operated using frequency modulation on six non-simultaneous channels. The radio signal was collected using a receiver mounted on the tractor to control the hydraulic system acting on the tractor wheels by means of suitable relays. In the event of power failure or a lack of radio signal, the fuel injection pump, via a suitable spring, was able to stop the fuel flow to turn off the engine. Light controls on a panel mounted onboard the tractor confirmed the right operation of the channels.

In the middle of 1957, still in England, a farmer mounted a transmitter on a Fordson tractor to manage a second identical tractor, suitably equipped with a receiver, relays and hydraulic control system, to reproduce the first tractor operations via a radio-controlled signal. A single operator could therefore operate two tractors simultaneously.

A third prototype proposed in England at the end of 1958 was a tractor that was able to automatically follow, by means of two sensing coils arranged symmetrically in the front part of the tractor, an electric pilot cable placed on the ground that was powered with a low-voltage alternating current. The signal generated in the coils acted on the hydraulic system through multi-stage relays, which were used to operate the steering wheel of the tractor by means of a double-acting cylinder to maintain the alignment with the pilot cable. When the tractor was in line with the cable, the signal from the coils was balanced.

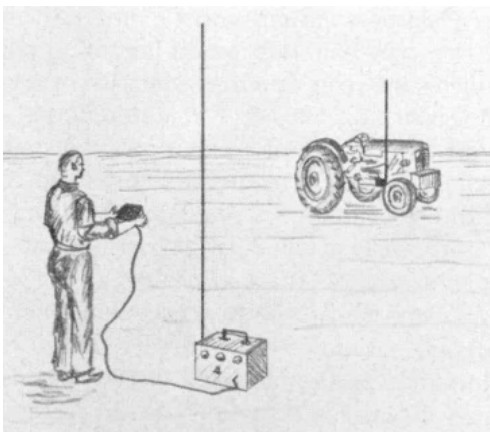

**Figure 8.** The Fordson radio-controlled tractor that was developed in England in 1955 [9].

A similar approach, but with different purposes, is dated at the end of 1958 in the United States. The system consisted of a mechanical device based on a metal feeler made of two elements and mounted on a Richey–Fordson 2WD tractor (Figure 9). During the operation, the feeler slides along the sides of the treated rows, and if the tractor deviated from the direction parallel to the row, one of the elements sensing the row caused a slight rotation of the feeler pivot to provide for the closure of one of the two contacts linked to the steering system. Consequently, through suitable relays, a double-acting hydraulic jack operating on the steering axle of the tractor was actuated to maintain the tractor parallel to the row of the crop. The system allowed for reducing the effort of the operator in terms of attention required to precisely drive but did not eliminate the need for an operator to ride on the tractor.

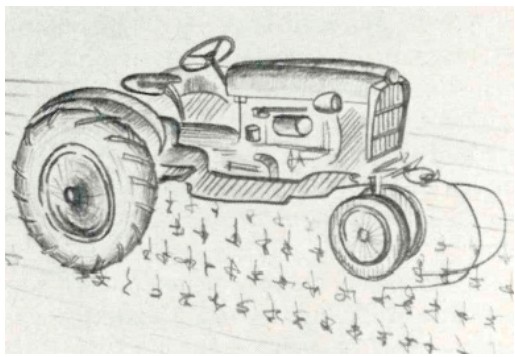

**Figure 9.** The Richey–Fordson self-driving tractor (1958) [9].

In 1959, news from Sweden related to a radio control device built and installed on a Bolinder Munktell tractor was released with no further information.

In the same year, information regarding a Caterpillar tractor called "S80" came from Russia (Figure 10). The tracklaying tractor was mounted with a system that allowed it to automatically follow a straight or slightly curved furrow that was previously traced on the soil by the driver. Nevertheless, at the two ends of the furrow, the tractor had to be manually driven and turned by the operator waiting for it. However, as reported by Loghinov (1959) [23], the first practical experience of the prototype was dated in 1957, when in 17 days, about 400 ha were plowed.

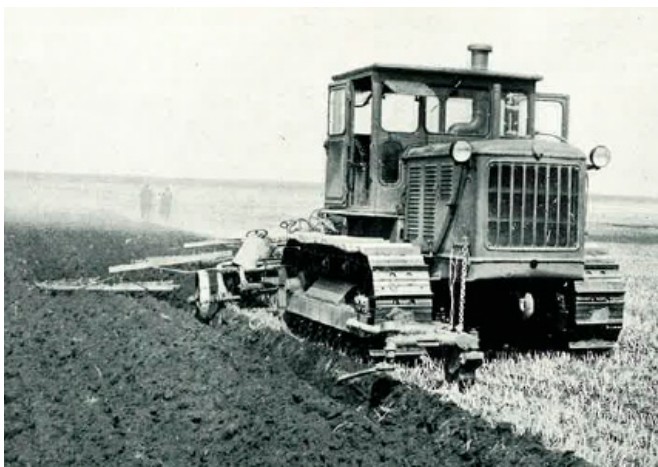

**Figure 10.** The tracklaying tractor that was designed in Russia in 1959 [23].

In 1960, some news of exhibitions of self-guided and radio-controlled tractors came from North America (University of Nebraska). In the case of autonomous tractors, the vehicle was able, without the driver's effort, to follow rows by means of feelers and actuators operating on the tractor steering organs. With respect to radio-controlled applications, the tractor was equipped with a radio receiving device that was remotely controlled by an operator via radio transmitting equipment.

To summarize, the interest in ground autonomous vehicles dates back to the 1950s, albeit with different design criteria; no one proposed a solution that was able to completely exclude human intervention while in operation.

### 3.1. Historical Contribution and Research from the University of Bologna

The Institute of Agricultural Mechanization (IMA) of the Alma Mater University of Bologna modified the ROSSI R4/35 tractor to obtain a fully autonomous vehicle for agriculture. The first step in the vehicle development was to design a remote radio-controlled tractor and then to update the version to the BOPS-1960 unmanned tractor.

The ROSSI tractor was an isodiametric four-wheel-drive tractor powered by a two-cylinder four-stroke diesel engine to provide a maximum power of 26 kW. The forward travel and the steering action were achieved by two pairs of brakes and clutches installed on the sides of the tractor similar to a tracklaying tractor. On the rear of the tractor, a hydraulic lift was provided for supporting a mounted plow. The first experimental activity was the evaluation of the performance and the features of the original tractor [24]. Tests were performed to analyze the performance at the power take-off (PTO) and the drawbar. Using a brake bench to connect the PTO shaft, the power, resistant torque and fuel consumption were measured.

Traction tests were carried out using a drawbar loading unit, mounting different tires and inflating pressure to consider the tractor behavior in varying soil conditions, such as heavy clay soil and farm roads. The forward speed, the towing performance, the power at the trailer hitch, the slippage and the efficiency at the trailer hitch were measured.

During the tests, the tractor was operated by means of a lever to engage and disengage the main clutch, two levers were used to actuate the steering clutches, a pedal lever was used to operate the steering brakes and, finally, a lever was used to control the hydraulic lifting device.

#### 3.1.1. The IMA 1959 First Radio-Controlled Tractor

The design of the first Italian prototype was in line with the Russian tractor implementation; nevertheless, at the time, the UNIBO research group ignored the activity of the Russian research team. The IMA 1959 radio-controlled tractor was presented during the open-field demonstration days in Ozzano Emilia, a small town in the Bologna province,

from 28 to 30 June 1959. The agricultural tractor was equipped with a remote-controlled system [25]. The tractor controls, originally consisting of manual levers, were modified or fully replaced with several electrical actuators. The maneuvers of the tractor were controlled by closing electrical circuits, both by means of buttons placed on a dashboard panel connected to the tractor via a multipolar cable and by means of a special radio-electric device (Figure 11).

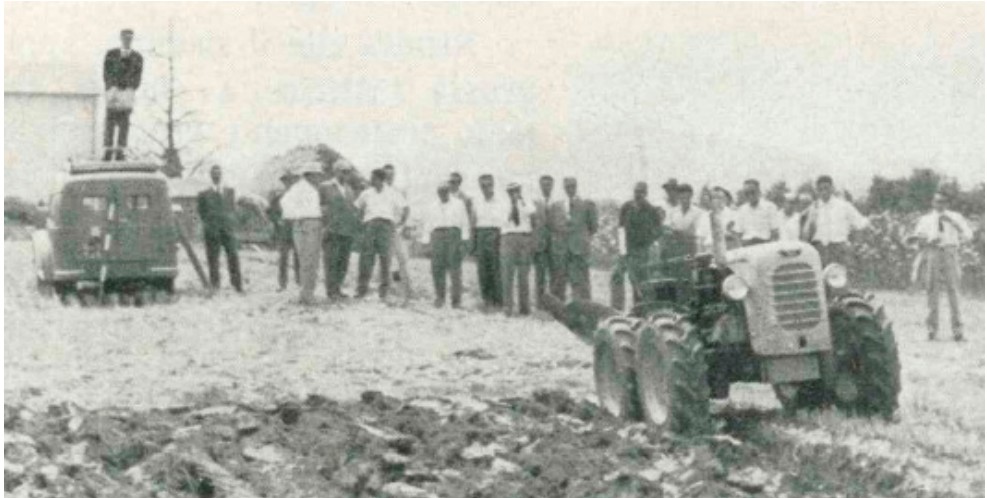

**Figure 11.** The 1959 IMA radio-operated tractor [9].

The implementation of the remote-control system of the tractor was based on reproducing the operations that are normally carried out manually by acting on the tractor levers and controls by means of hydraulic pistons operated by valves, which were actuated by electromagnets.

In the tractor, a complex device consisting of three separated systems was installed:

a. Auxiliary hydraulic system;
b. Auxiliary electrical system (chain of relays);
c. On-board radio-receiver system.

Compared to the Russian prototype, the vehicle was more complete with respect to the implemented controls; indeed, all the operations involved with soil plowing, including the turns at the end of the field and the empty return, were remotely performed.

3.1.2. BOPS-1960: The Unmanned Programmable Tractor

In 1960, the version of the prototype, defined as the "BOPS 1960", unmanned programmable tractor (Figure 12), was developed and presented during the agricultural field events in Pesaro, Torrette di Fano, 26–29 June 1960.

This agricultural tractor prototype was designed to be capable of operating completely autonomously according to a preset program [9]. Toward this aim, the auxiliary hydraulic system was replaced by small electric motors powered by onboard batteries. Thus, all the onboard actuators acted by means of electrical contacts. The installed electric motors were used to control the following:

a. The main clutch;
b. The steering brakes and clutches (Figure 13a);
c. The throttle;
d. The hydraulic lifting device (Figure 13b).

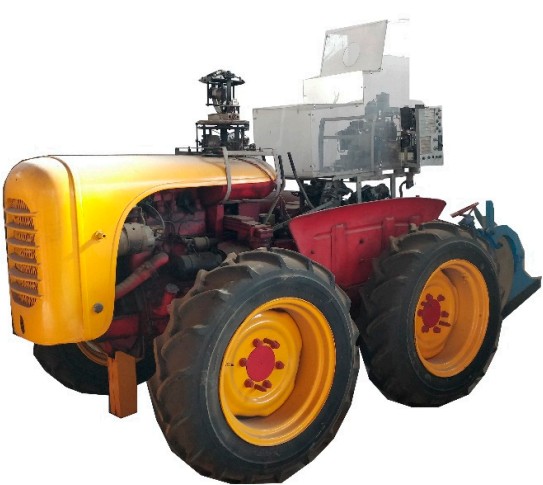

**Figure 12.** BOPS-1960: the unmanned programmable tractor.

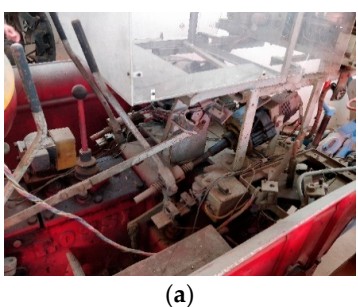

(**a**)

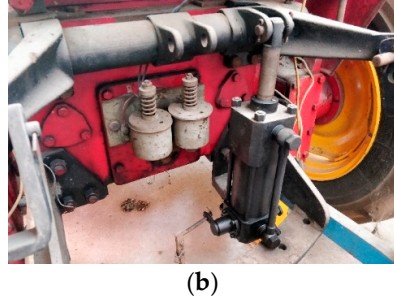

(**b**)

**Figure 13.** (**a**) Detail of the electric motor connected to the left clutch lever and (**b**) electromagnets to control the hydraulic lift of the tractor.

In addition, a new system called the programmer unit was installed (Figure 14). Its purpose was to determine, at the appropriate time, the commands to the vehicle according to its configuration.

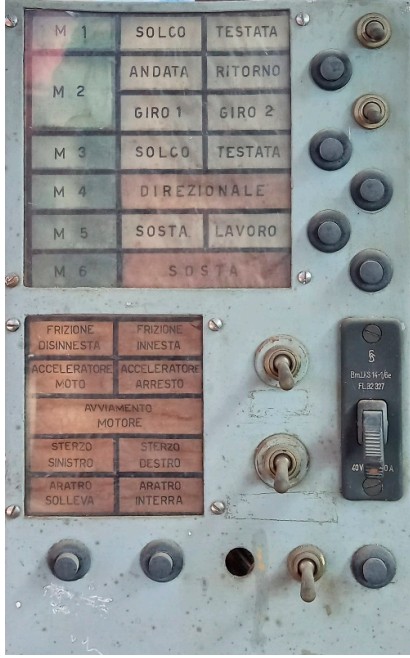

**Figure 14.** Control panel of the programmer unit mounted on the BOPS-1960.

Two other auxiliary units, which were mechanically independent but operating in close electrical connection, were installed on the vehicle: an inertial "directional unit" and a mechanical feeler that was mounted on the front side of the tractor. The logic operation was implemented so that after manually starting the tractor, the programmer unit controlled the directional unit and the electrical motors to actuate the mechanical controls.

The directional group [26] was composed of a mechanical gyroscope with three degrees of freedom (Figure 15). The gyroscope acts to maintain the straight direction of the tractor and to perform 90° turning maneuvers using a rotation of the gyroscope via an electric motor.

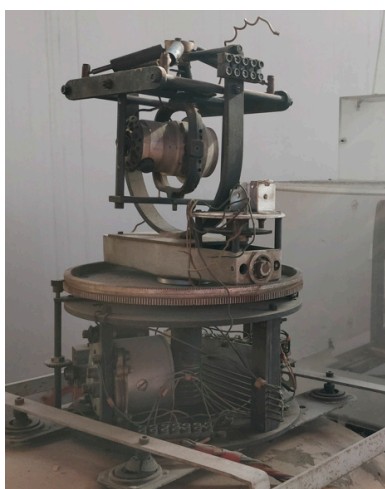

**Figure 15.** The directional unit composed of the mechanical gyroscope with three degrees of freedom and mounted on a base frame on the BOPS-1960.

The feeler unit [27] was composed of a rod extending in the forward direction, ahead of the right wheel. Naturally, this lay on the ground, but it was free to move both in the vertical and the horizontal planes. The two degrees of freedom of the feeler allowed for controlling the direction of motion and the starting/stopping of the plowing operation. In fact, the feeler during the plowing operation could touch the bottom of the furrow opened in the previous operation and help to correct the direction of motion until the end of the furrow. In this location, the lifting of the rod acted on the hydraulic lift to raise the plow and therefore stop the plowing. As it was implemented, the system required the two initial furrows to be preliminarily traced on both sides of the field to be able to correctly perform the plowing operation. The two initial furrows served as a guide for the subsequent furrows.

The programmer unit [26] consisted of electric motors to power a system consisting of circular camshafts (Figure 16), which operated above electrical contacts. While rotating, they enabled, at the appropriate time, different tractor control inputs according to the predetermined sequence. Each cam managed two operations in sequence. The complete revolution of the cam corresponded to two separate operations. The time of the two operations was determined by the geometry of the cam and the rotation speed of the electric motor, which was computed by using times greater than the design ones. Then, the operations followed at determined time intervals. The unavoidable variability of the operating times that occur while executing the plowing tasks in the field had to be considered to guarantee the right flexibility of the operations. The issue was solved by coupling a second motor to the cam system. The second motor was activated by the signals coming from the directional group or the feeler group, thus causing the camshaft to rotate faster, effectively speeding up the operation until the next operation was started.

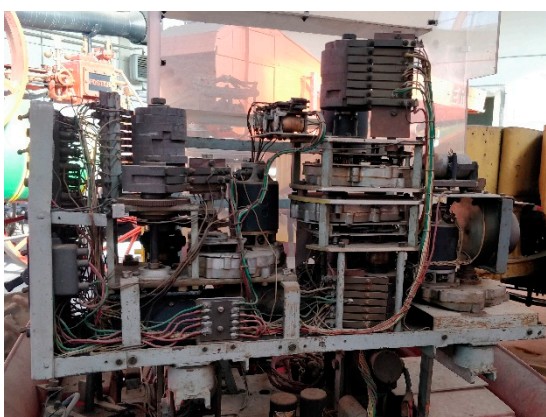

**Figure 16.** BOPS-1960 programmer unit. The circular camshafts are clearly visible.

## 4. Conclusions

Objective differences between the autonomous machines of the past and the current prototypes are a matter of fact. However, the early prototypes were derived from the tractors on the market. Conversely, current UGVs are often specifically designed units with defined shapes and dimensions.

A consequence of this is that while tractors are extremely versatile vehicles, the modern UGVs are often specialized machines. The vehicles and platforms reviewed were described based on the operation performed. The DEDALO from the University of Bologna, equipped with a sprayer and a shredder, and Armadillo, the Danish platform that was able to be a sprayer unit or a robot for scouting or crop monitoring, were recorded as the most versatile units. Nevertheless, the UNIBO research team is currently working on expanding the performance of the prototype by designing new implements to increase the autonomous farming tasks for precision orchard and vineyard management and to replace the combustion engine with an electric one.

Nowadays, the research contributions in Europe to the development of the modern autonomous tractors are less active relative to the attention given in the past. This is mainly due to autonomous tractors already potentially being available using the current commercial technology; nevertheless, because the European regulation does not allow for fully autonomous tractors, it is not marketable yet. In effect, tractors with the most advanced driving systems on the European market nowadays are the vehicles with a semi-autonomous driving system that is able to carry out any field operation with full autonomy, including turning at the end of the field, but the operator has to remain in the driver's seat to directly drive the vehicle if necessary.

Furthermore, the current industry trend is often to manufacture larger and larger tractors with a higher power, often leading to problems related to soil compaction. Consequently, the smaller, lighter, and less powerful modern UGVs represent an interesting alternative in terms of environmental sustainability. Given the low power of the unmanned platforms, it is predictable that they are efficient for light operations unless some new generation implements will be designed.

A future scenario for autonomous tractors and light UGVs will be to have separate markets but soon they will coexist in normal farming operations.

Sixty years ago, autonomous tractors performed repetitive actions by relying exclusively on ground positioning systems. Nowadays, fully autonomous vehicles are at the very final stage of prototyping, and the current technologies available, such as the global positioning systems and on-the-go cameras, allow for operating in-field tasks in real time to adapt to the variable conditions of the working area.

Solutions and devices that were unthinkable sixty years ago are now of daily use. It is therefore natural to wonder how the autonomous vehicles of the future will evolve in relation to the fast technological progress we are witnessing today. Nevertheless, it is

always important to know the findings of the research performed in the past to be able to progress by enhancing the actual contributions by avoiding duplicating the work and mainly the mistakes of the past by gaining experience and knowledge.

**Author Contributions:** Conceptualization, V.R. and B.F.; methodology, V.R. and B.F.; writing—original draft preparation, V.R. and B.F.; writing—review and editing, V.R., B.F. and D.M.; supervision, V.R. and B.F.; funding acquisition, V.R. All authors have read and agreed to the published version of the manuscript.

**Funding:** This work was funded by the Italian MIUR within the project "New technical and operative solutions for the use of drones in Agriculture 4.0" (PRIN 2017, Prot. 2017S559BB).

**Data Availability Statement:** Not applicable.

**Acknowledgments:** The authors appreciate the support and help of Giulia Piovaccari in references analysis.

**Conflicts of Interest:** The authors declare no conflict of interest.

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
