# Peer review of "A Review of Current and Historical Research Contributions to the Development of Ground Autonomous Vehicles for Agriculture"

_sustainability, doi:10.3390/su14159221_

Round 1
Reviewer 1 Report
The current research is very good review about the use of autonomous vehicles in Precision Agriculture. I found this paper very interesting where Several technical aspects were nicely implemented and explained sufficiently. I am sure, authors invested huge amount of time and have made a great effort to produce this high-quality of research which is clearly structured and the language used is largely appropriate.
â–ºThe title of the paper looks great.
· â–ºI suggest for the authors to rephrase the abstract as a whole and present it a more attractive way.
· â–º The list of key words must extend to include at least five words in direct relation to the theme.
· â–ºPlease introduce the whole content of the paper in the last paragraph of the introduction.
· â–º All the acronyms form their first appearance in the paper.
· â–º Also, all the references MUST BE CHECKED and formatted as required by MDPI-SUSTAINABILITY also make sure that all the references have DOI number unless it is not available.
Author Response
Response to Reviewer 1 Comments
Dear reviewers:
We wish to express our thanks for your careful comments on our manuscript entitled “A review of current and historical research contributions to the development of ground autonomous vehicles for agriculture” (sustainability-1746152). The comments were very valuable and helped us to improve the manuscript. Revisions are marked in red in the manuscript.
Please find below our point-by-point responses to the reviewer comments:
Specific comments
The title of the paper looks great.
Response: Thank you very much
I suggest for the authors to rephrase the abstract as a whole and present it a more attractive way.
Response: Revised accordingly.
The list of key words must extend to include at least five words in direct relation to the theme.
Response: Revising accordingly, a fifth keyword has been added..
Please introduce the whole content of the paper in the last paragraph of the introduction.
Response: Revised accordingly.
All the acronyms form their first appearance in the paper.
Response: Revised accordingly.
Also, all the references MUST BE CHECKED and formatted as required by MDPI-SUSTAINABILITY also make sure that all the references have DOI number unless it is not available.
Response: Revised accordingly the sustainability format.
Reviewer 2 Report
Dear Author,
You can improve the paper with some aspects:
What is the relevance of yours study? Why it was done? What was the purpose of this review of autonomous vehicles for agriculture?
The paper is a review of some agricultural equipment with certain technical characteristics.
It should have been highlighted, possibly, their role in the further development of different types of such equipment, to specify what is their efficiency/performance in carrying out agricultural works.
You can summarize this information by highlighting how these devices / technological innovations have been integrated into the mass production of agricultural equipment.
Regards,
Author Response
Response to Reviewer 2 Comments
Dear reviewer:
We wish to express our thanks for your careful comments on our manuscript entitled “A review of current and historical research contributions to the development of ground autonomous vehicles for agriculture” (sustainability-1746152). The comments were very valuable to improve the manuscript. Revisions are marked in red in the manuscript.
Please find below our point-by-point responses to the reviewer comments:
Specific comments
What is the relevance of yours study? Why it was done? What was the purpose of this review of autonomous vehicles for agriculture?
Response: The authors idea was to describe the research contributions in the design and prototyping of agricultural ground unmanned vehicles from the first UGV conceived in the 60s of the last century to the current day and this point is in the manuscript.
The paper is a review of some agricultural equipment with certain technical characteristics.
Response: yes but with attention to UGV prototypes developed in research centers.
It should have been highlighted, possibly, their role in the further development of different types of such equipment, to specify what is their efficiency/performance in carrying out agricultural works.
You can summarize this information by highlighting how these devices / technological innovations have been integrated into the mass production of agricultural equipment.
Response: In the conclusions we specify the role of these vehicles and the current limitation in the European market to the development of unmanned agricultural tractors in reason of the current regulation.
Reviewer 3 Report
The research is limited to the historical evolution of ground autonomous vehicles for agriculture without providing scientific results and conclusions.
This paper can be considered a chapter of a specialized book.
We recommend that authors identify research directions that lead to these results.
The authors do not present the research methodology, they have not formulated hypotheses, and they do not manage to substantiate empirical research.
the historical presentation can be accompanied by the bibliometric analysis of the scientific papers that have documented this topic
Author Response
Response to Reviewer 3 Comments
Dear reviewer:
We wish to express our thanks for your comments on our manuscript entitled “A review of current and historical research contributions to the development of ground autonomous vehicles for agriculture” (sustainability-1746152). The comments helped us to improve the manuscript. Revisions are marked in red in the manuscript.
Please find below our point-by-point responses to the reviewer comments:
Specific comments
The research is limited to the historical evolution of ground autonomous vehicles for agriculture without providing scientific results and conclusions.
Response: the manuscript is a review paper to highlight the contribute of research centers over the years to the development of UGV for agriculture
This paper can be considered a chapter of a specialized book.
Response: Thank you for your suggestion. In future time we can consider the suggestion.
We recommend that authors identify research directions that lead to these results.
Response: The authors idea was to describe the research contributions in the design and prototyping of agricultural ground unmanned vehicles from the first UGV conceived in the 60s of the last century to the current day.
The authors do not present the research methodology, they have not formulated hypotheses, and they do not manage to substantiate empirical research.
the historical presentation can be accompanied by the bibliometric analysis of the scientific papers that have documented this topic.
Response: We have tried to address the manuscript as a review paper.